# Relationship between Perceived Indoor Temperature and Self-Reported Risk for Frailty among Community-Dwelling Older People

**DOI:** 10.3390/ijerph16040613

**Published:** 2019-02-20

**Authors:** Yukie Nakajima, Steven M. Schmidt, Agneta Malmgren Fänge, Mari Ono, Toshiharu Ikaga

**Affiliations:** 1School of Science for Open and Environmental Systems, Graduate School of Science and Technology, Keio University, Hiyoshi 3 14 1, Kohoku, Yokohama, Kanagawa 2238522, Japan; steven.schmidt@med.lu.se (S.M.S); onomari@a6.keio.jp (M.O.); 2Japan Society for the Promotion of Science, Koujimachi 5 3 1, Chiyoda, Tokyo 1020083, Japan; 3Department of Health Sciences, Faculty of Medicine, Lund University, Box 157, 22100 Lund, Sweden; agneta.malmgren_fange@med.lu.se; 4Department of System Design Engineering, Faculty of Science and Technology, Keio University, Hiyoshi 3 14 1, Kohoku, Yokohama, Kanagawa 2238522, Japan; ikaga@sd.keio.ac.jp

**Keywords:** home, old age, winter season, economic satisfaction, fall risk

## Abstract

This study investigated the relationship between perceived indoor temperature in winter and frailty among community-dwelling older people. This cross-sectional study included 342 people 65 years and older in Japan. Participants answered questions about demographics, frailty, housing, and perceived indoor temperature in winter. Participants were grouped based on perceived indoor temperature (Cold or Warm) and economic satisfaction (Unsatisfied or Satisfied). Differences in the frailty index between perceived indoor temperature groups and economic satisfaction groups were tested by using ANCOVA and MANCOVA. An interaction effect showed that people in the Cold Group and unsatisfied with their economic status had significantly higher frailty index scores (*F*(1, 336) = 5.95, *p* = 0.015). Furthermore, the frailty index subscale of fall risk was the specific indicator of frailty that accounted for this significant relationship. While previous research has shown the risks related to cold indoor temperature in homes, interestingly among those who reported cold homes, only those who were not satisfied with their economic situation reported being at increased risk for frailty. This highlights the potential importance of preventing fuel poverty to prevent frailty.

## 1. Introduction

Recently, the impact of cold weather on functioning and health has attracted attention. Cold weather is associated with lower physical activity [1], increased blood pressure and excess cardiovascular mortality, asthma and respiratory symptoms [2]. Falling temperature and colder weather are also associated with increased risk of aneurysmal subarachnoid hemorrhage [3], and to various negative functional outcomes [4]. From an experimental study using a movement laboratory in a climate chamber, Lindemann et al. [5] reported that the physical performance of older women exposed to cold temperature (15 °C) was significantly lower than women exposed to warm/normal temperature (25 °C). Additionally, seasonal trends in the physical performance of older people were associated with a lower level of performance in the winter compared with the autumn, and people living in colder houses had a lower level of physical performance than those living in warm houses [6]. With advancing age, the efficiency of adaptive mechanisms to regulate temperature declines, and thus the comfort zone for older people is stated at higher ambient temperatures than in their younger counterparts [7]. Perception of cold among older people has been found to be associated with poor sleep quality [8] and poor self-reported health [9].

With increasing age and health decline, people spend more time doing activities in their homes [10,11], and thus it is crucial to understand what role the indoor temperature in the home plays for their health. In the Cold Weather Plan for England [12], older people, in particular those who are frail or socially isolated and over 75 years of age, are considered to be particularly at-risk in the event of severe cold weather. Overall, the problem of cold weather for older people is framed as an integral problem of lower physical functioning and performance of activities of daily living (ADL), energy inefficient homes and heating systems, financial poverty, and old- fashioned attitudes such as partial intermittent heating [13]. 

Frailty is widely recognized as a considerable challenge for the society, but there is no single agreed definition or cause of the phenomenon. The definition suggested by Fried et al. [14] includes self-reported exhaustion, reduced grip strength, slow walking speed, and low level of physical activity. In addition to physical functioning, it has been suggested that definitions of frailty should also include aspects of mental health, such as cognition and mood [15]. Shinkai et al. [16,17] include three domains in their definition of frailty: isolation risk, fall risk, and nutrition risk. With increasing age, frailty becomes more frequent and severe [18,19,20,21,22]. Frailty is associated with chronic disease [23], obesity [24], and female gender [20,25,26], as well as low education level and income [14]. When it comes to the impact of the environment, it is well known that being confined to the home is a risk factor for walking limitation and declining ADL [27,28], thus increasing the risks of falling [29,30] and subsequent mortality [31]. That is, different conditions in the environment may well contribute to worsen the consequences of frailty. Frailty most often leads to limitations in ADL, ultimately requiring nursing home placement [32] and increased use of health care resources [33]. For informal caregivers, care-recipient’s frailty can be a significant predictor of caregiver burden [34,35], which may lead to emotional distress [36], poor health, decreased quality of life [37] and increased health care consumption [38].

There are very few studies on the relationship between indoor temperature and frailty, and we are not aware of any studies focusing on the relationship between perceived indoor temperature and frailty. Hence, the overarching aim of this study was to investigate the relationship between perceived indoor temperature and frailty among older people living in the community. The specific aims are to: (1) investigate the relationships between self-reported risk for frailty and perceived indoor temperature, and (2) to determine whether economic satisfaction influences the relationship between self-reported risk for frailty and perceived indoor temperature.

## 2. Materials and Methods

### 2.1. Procedure and Participants

Data collection for this cross-sectional study was conducted during December of 2014, 2015, and 2017. Staff at outpatient rehabilitation facilities in Kochi, Osaka, and Yamanashi prefectures, Japan provided surveys to their clients. The surveys were either completed at the rehabilitation facility or at the participant’s home, and 473 were returned. To be included in this study, participants had to be 65 years or older and use the facilities one to two times per week for physical rehabilitation. The study protocol was approved by the Keio University Ethics Review Board on 4 August 2014 (26-11), 29 July 2015 (27-31), and 29 August 2017 (29-79). All participants received oral and written information, and the questionnaires were filled out anonymously. Of the 473 surveys returned, 131 were excluded: 10 participants were younger than 65 years and 54 participants did not disclose their age. Additionally, 67 participants did not complete questions from the frailty index and/or perceived indoor temperature. Hence, 342 people were included in the present study. Participants included in this study did not differ from those who were excluded on any characteristic except for the score on poor nutrition risk which is one of the subscales of frailty index. Mean scores of poor nutrition risk were 0.97 (SD = 0.96) for the included participants and 0.73 (SD = 0.86) for the excluded participants (*p* = 0.030).

### 2.2. Measurements

#### 2.2.1. Participant Characteristics

Participants’ characteristics included self- reported age, gender, body mass index (BMI), level of education (junior high school or less/senior high school/university or higher), and whether the person lived alone (yes/no). 

#### 2.2.2. Frailty Index

To measure frailty, we used the “Kaigo-Yobo Check-List” proposed by Shinkai et al. [16,17], which can be easily self-administered by responding to 15 items (Table 1). All of the items are answered on a two-point scale, positive answer (0) or negative answer (1). The items are summed to calculate a total score ranging from 0 to 15 with higher scores indicating a higher risk of frailty. This checklist has three sub-scales that measure the risk of becoming frail: isolation risk, fall risk, and nutrition risk. Sum scores can also be calculated for each subscale: Isolation Risk (0–5), Fall Risk (0–6), and Nutrition Risk (0–4).

#### 2.2.3. Perceived Indoor Temperature

Perceived indoor temperature was examined using the standardized questionnaire, Comprehensive Assessment System for Built Environment Efficiency (CASBEE) Health Checklist [39]. The checklist was created to help residents determine whether or not environmental factors and housing equipment have any potential risks for their health. It consists of questions about the levels of heat in summer, cold in winter, noise, light, cleanliness, safety, and security of the living environment for each room in the house. Respondents are asked to mark the frequency of corresponding problems on a four-pointed scale: 1) often (0 point), 2) occasionally (1 point), 3) rarely (2 point), 4) not at all (3 point). The scores for each item are summed and higher scores indicate fewer problems in the living environment. In this study, six questions about feeling cold in the living room, bedroom, dressing room, bathroom, toilet, and corridor were used to evaluate the perceived indoor temperature in winter. The scores on perceived indoor temperature from the CASBEE Health Checklist were not normally distributed. Therefore, participants with higher scores (10–18 points) were classified into the Warm Group, and participants with lower scores (0–9 points) were classified into the Cold Group.

#### 2.2.4. Economic Satisfaction

Economic satisfaction was measured with a study specific question, with four response choices: “Very satisfied”, “Somehow satisfied”, “Not very satisfied” and “Not satisfied at all”. Economic satisfaction was dichotomized for the analyses: “Very satisfied” and “Somehow satisfied” were considered as “Satisfied”, and “Not very satisfied” and “Not satisfied at all” were considered as “Unsatisfied”.

### 2.3. Data Analysis

Analysis of covariance (ANCOVA) was used to test for differences between the perceived indoor temperature groups and the economic satisfaction groups with the frailty index as the dependent variable. Perceived indoor temperature, economic satisfaction, and the interaction between these two factors were in the model as independent variables. Since we know from previous research that they are associated with frailty, gender, living alone or not, education, age and BMI were in the model as covariates. Spearman’s correlations between each covariate and the independent variables were not larger than 0.2. Multivariate analysis of covariance (MANCOVA) was used to test the three subscales of the frailty index (isolation risk, fall risk, and nutrition risk) using the same independent variables and covariates as the previous model. Wilks’ Lambda was used as the omnibus test of the MANCOVA. Both models used the general linear model for the analyses. *p*-value < 0.05 was considered statistically significant. All statistical analyses were performed with SPSS version 24.0 (IBM, Armonk, NY, USA).

## 3. Results

Among the 342 participants included in the analysis, the mean age was 81.74 (SD = 7.27) years, and 215 (62.9%) were women (see Table 2 for more details).

In the ANCOVA for total score of frailty index with the perceived temperature and economic satisfaction groups (Table 3), there was a significant interaction effect, *F*(1, 328) = 5.28, *p* = 0.022. The levels of reported frailty risk were not significantly different, *F*(1, 328) = 0.04, *p* = 0.833, between the Warm Group and Cold Group among those participants who were satisfied with their economic status. In contrast, the levels of frailty risk were significantly higher, *F*(1, 328) = 7.33, *p* = 0.007, among those in the Cold Group who were also unsatisfied with their economic status (Figure 1).

The omnibus test of the MANCOVA was significant with Wilks’ Lambda = 0.98, *F*(3, 326) = 2.78, *p* = 0.041. The main effects and interaction effects for isolation risk and nutrition risk were not significant, but there were significant differences in the fall risk subscale (Table 4). The Warm Group and Cold Group were not significantly different, *F*(1, 328) = 0.31, *p* = 0.576, on the fall risk subscale among those participants who were satisfied with their economic status. In contrast, the Cold Group scored significantly higher on the fall risk subscale, *F*(1, 328) = 12.15, *p* = 0.001, among those who were also unsatisfied with their economic status (Figure 2).

## 4. Discussion

In the present study, we found that those who were unsatisfied with their economic situation and also perceived the indoor temperature in their home as cold reported a higher risk of frailty than other groups. In contrast, people who were satisfied with their economic status reported similar frailty risks regardless of perceived indoor temperature. We also found that the reported fall risk had the strongest relationship with perceived temperature and economic satisfaction compared with nutrition and isolation risks. 

The most interesting finding was the interaction between economic satisfaction and reported frailty risk. We found that the people who both perceived a cold indoor winter temperature and were unsatisfied with their economic situation reported the highest risk of frailty. This same pattern was also seen for the specific indicator of fall risks but not for nutrition risk or isolation risk. Hence, the total frailty risk in our sample seems to be accounted for by the fall risk. Previous research found that poverty risk was associated with increased levels of frailty, and rather than educational or behavioral factors, material and in particular, psychosocial factors such as perceived control and social isolation explained a large part of poverty-risk-related differences in frailty [40]. To be more specific, it has been reported that low income is associated with under-nutrition [41] and loneliness or depression in the elderly [42], but those relationships did not seem to hold in our study in relation to frailty risk. When it came to people who were satisfied with their economic situation, there was no difference in reported frailty risk regardless of perceived indoor temperature. As previously reported, higher income seems to reduce the risk for frailty as those with higher incomes are more likely to survive into old age through better health status [43]. 

Hence, we found that it was not just lower economic status that was related to frailty risk, but it was the specific group that also reported living in a colder house that had the strongest relationship with frailty risk. This finding argues for encouraging heating up the homes of older people as a means to reduce risk of frailty and potentially promote healthy ageing. Specifically, different approaches to frailty prevention may be necessary for different subgroups. One possibility was that people unsatisfied with their economic situation were the people having difficulty warming up their house [44] and at risk of fuel poverty. In fact, older people are at great risk from fuel poverty, since they are more likely to be retired or on fixed incomes [45]. This study highlights the potential importance of preventing fuel poverty to prevent frailty and suggests that health promotion strategies must include low- income older people as a target group to improve their housing by installing proper insulation and heating systems. 

Out of the three specific indicators of frailty, fall risks had the strongest association with perceived indoor temperature and economic satisfaction. That is, those with poor economic status and perceived cold homes had a higher fall risks, while those satisfied with their economic status had a lower fall risks regardless of perceived indoor temperature. The result that fall risks was the only specific indicator of frailty with a significant association with perceived indoor temperature was expected and in line with previous findings. For example, we have reported physical performance decline among older people due to cold season and cold indoor temperature [6], and similar findings have also been reported by Lindemann et al. [5]. Yeung et al. [46] reported that there are higher incidences of falls in winter than in other seasons. Additionally, they highlighted that a higher proportion of fallers during winter had lower limb weakness than those who fell in non-winter seasons [46], as also indicated in our study.

From a health perspective, there is much to win from reducing the risks of frailty, and to prevent physical decline and subsequent falls. Regular physical activity has strong effects on reducing risk of premature death and chronic diseases [47,48]. Ultimately, the prevention of frailty improves perceived health since physical health, mental health and participation in physical activities all contribute to quality of life [49]. Furthermore, there are benefits for the society such as reduced needs for health care and social services and thus, the public expenditure.

When it comes to methodology, the current study has some limitations. First, due to the cross-sectional design we could not conduct an assessment of causality. Exploring casual association is important since it may play an instrumental role in terms of identifying reasons behind a wide range of processes, as well as assessing the impacts of changes on existing norms or processes. However, in order to study causal associations, follow-up studies are required. Second, the participants were not randomly selected. Participants included in the analysis had slightly higher risk of poor nutrition than those who were excluded. However, the rehabilitation facilities were located in different parts of Japan, and the age distribution of the participants was similar to other Japanese studies targeting frail elderly people living in the community and using rehabilitation services [50,51,52]. Hence, we are confident that the results represent the sub-population receiving rehabilitation services. Third, we assessed indoor temperature based on individual perception rather than directly measuring the temperature using thermometers. It has been reported that thermal sensitivity declines with ageing [53] and the perceived indoor temperature might not reflect the actual indoor temperature in all cases. This concern becomes larger especially in this study because all participants were frail older people, and their thermal sensitivity might be less sensitive than among healthier older people. However, when screening large populations, measurement-based investigations can be difficult to conduct because they are time consuming and costly and often require trained investigators’ support for installing thermometers in participants’ houses due to reduced independence of participants. Finally, we chose not to ask about actual income; instead we used economic satisfaction as an indicator of economic status. In fact, actual income may not have been a better measure, as the structure of a household in Japan can be very complex. In three-generation households, the employment income of the younger generations is likely to be the main household income, whereas in elderly one-person households, a public pension is likely to provide the main income [54]. Therefore, the economic status of older people changes with household structure and level of public pension.

## 5. Conclusions

Our results support the relationship of cold indoor temperature in the homes of older people with frailty risk. Our findings add a new contribution to studies related to frailty among older people living in the community by including the relationship with perceived indoor temperature and economic satisfaction. We can conclude that, from a public health perspective, there would be benefits from programs to support older people to maintain a warm indoor temperature: for example, by supplementing heating costs or retrofitting homes with improved insulation and multi-pane windows. For older people and their families, such efforts would probably lead to declining risks for falls and ultimately to maintenance or improvement of health. For society, such measures would reduce the needs for health care and social services and potentially reduce public expenditure.

## Figures and Tables

**Figure 1 ijerph-16-00613-f001:**
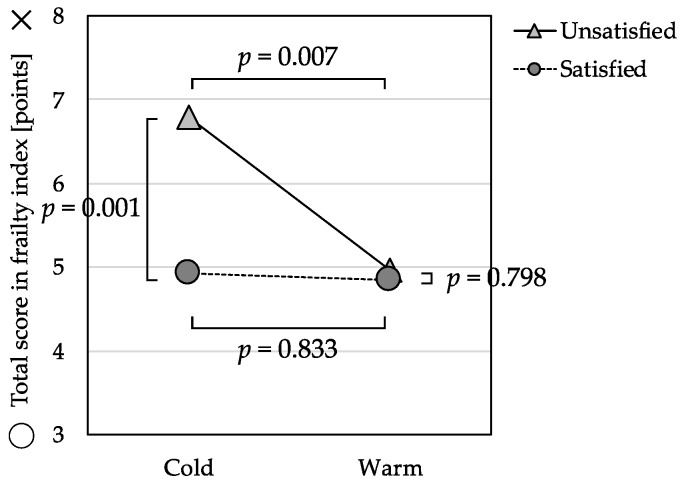
Interaction effect between perceived indoor temperature and economic satisfaction on the total frailty score (*n* = 342).

**Figure 2 ijerph-16-00613-f002:**
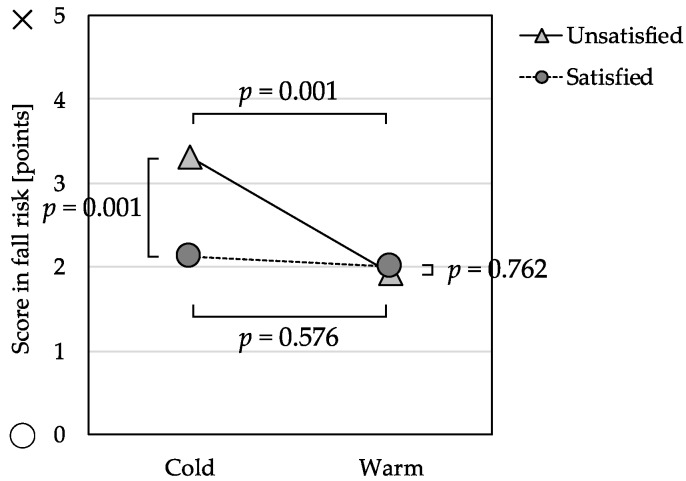
Interaction effect between perceived indoor temperature and economic satisfaction and its relation to fall risk score (*n* = 342).

**Table 1 ijerph-16-00613-t001:** The frailty index translated from Kaigo-Yobo Check-List [16,17].

Category	Item
Isolation risk	1) Do you often stay at home and do not go outside for a day? Positive answer: No / Negative answer: Yes
	2) How often do you go outside for work (including farming), shopping, walking, or hospital visits? *Does not include gardening or taking out the garbage. P: More than once in a couple of days / N: Less than once a week
	3) Do you have hobby(ies) in the house or outside the house? P: Yes / N: No
	4) Do you have friend(s) in the neighborhood? P: Yes / N: No
	5) Do you have friend(s) other than your neighbors, family or relatives living apart who keep in touch? P: Yes / N: No
Fall risk	6) Have you had a fall within the past one year? P: No / N: Yes
	7) Can you walk 1 km continuously? P: Can do it without any discomfort / N: Can do it, but with discomfort or cannot do it
	8) Do you have good eyesight? *You can use your glasses P: Yes, I can read a book / N: No, I cannot see well or cannot see anything
	9) Do you often slip or stumble in the house? P: No / N: Yes
	10) Do you avoid going outside because of fear of falling? P: No / N: Yes
	11) Have you been hospitalized in the past year? P: No / N: Yes
Nutrition risk	12) Do you have an appetite recently? P: Yes / N: No
	13) How much can you chew now? *You can use the artificial tooth P: I can chew most things / N: I cannot really chew and things to eat are limited
	14) Have you lost more than 3 kg of your weight in the past 6 months? P: No / N: Yes
	15) Do you think that you have lost more muscle or fat than before in the past 6 months? P: No / N: Yes

**Table 2 ijerph-16-00613-t002:** Participant characteristics by perceived indoor temperature and for total sample (*n* = 342).

	Total (*n* = 342)		Cold Group (*n* = 107)		Warm Group (*n* = 235)	
Age (years), mean (SD)	81.74	(7.27)	80.01	(7.51)	82.53	(7.03)
Female, *n* (%)	215	(62.9)	61	(57.0)	154	(65.5)
BMI, *n* (%)						
Underweight (<18.5)	46	(13.5)	17	(15.9)	29	(12.3)
Normal (18.5 ≤ BMI < 25)	192	(56.1)	61	(57.0)	131	(55.7)
Obese (≤25)	62	(18.1)	17	(15.9)	45	(19.1)
No answer	42	(12.3)	12	(11.2)	30	(12.8)
Economic satisfaction, *n* (%)						
Satisfied	63	(18.4)	28	(26.2)	35	(14.9)
Not satisfied	279	(81.6)	79	(73.8)	200	(85.1)
Education, *n* (%)						
Junior high school or less	29	(8.5)	16	(15.0)	13	(5.5)
Senior high school	134	(39.2)	41	(38.3)	93	(39.6)
University or higher	130	(38.0)	41	(38.3)	89	(37.9)
No answer	49	(14.3)	9	(8.4)	40	(17.0)
Family member, *n* (%)						
Living alone	80	(23.4)	18	(16.8)	62	(26.4)
Living with someone	255	(74.6)	87	(81.3)	168	(71.5)
No answer	7	(2.0)	2	(1.9)	5	(2.1)
Frailty index, mean (SD)						
Total score	5.01	(2.7)	5.51	(2.9)	4.79	(2.58)
Isolation risk	1.91	(1.34)	2.02	(1.44)	1.86	(1.3)
Fall risk	2.14	(1.56)	2.49	(1.72)	1.98	(1.46)
Poor nutrition risk	0.97	(0.96)	1.01	(1.04)	0.95	(0.92)

**Table 3 ijerph-16-00613-t003:** ANCOVA of total frailty score by perceived indoor temperature and participant characteristics (*n* = 342).

	*df*	Total Score in Frailty Index	
		*F*	*η* ^2^
Perceived indoor temperature	1	6.17 *	0.018
Economic satisfaction	1	7.01 **	0.021
Perceived indoor temperature * Economic satisfaction	1	5.28 *	0.016
Gender	1	0.28	0.001
Age	1	1.86	0.006
BMI			
Underweight vs Normal (A)	1	8.96 **	0.027
Obese vs No answer (B)	1	1.95	0.006
(A) vs (B)	1	2.58	0.008
Education			
Junior high school or less vs Senior high school (C)	1	0.97	0.003
University or higher vs No answer (D)	1	2.52	0.008
(C) vs (D)	1	2.18	0.007
Living alone			
Living alone vs Living with someone (E)	1	5.81 *	0.017
(E) vs No answer	1	0.02	0.000

* *p* < 0.05, ** *p* < 0.01, *η*^2^ Eta-squared, Adjusted *R*^2^ = 0.077.

**Table 4 ijerph-16-00613-t004:** Multivariate analysis of covariance of three subscales of the frailty index by perceived indoor temperature and participant characteristics (*n* = 342).

	*df*	Isolation Risk		Fall Risk		Nutrition Risk	
		*F*	*η* ^2^	*F*	*η* ^2^	*F*	*η* ^2^
Perceived indoor temperature	1	0.52	0.002	11.10 **	0.033	0.20	0.001
Economic satisfaction	1	1.53	0.005	6.04 **	0.018	2.63	0.008
Perceived indoor temperature * Economic satisfaction	1	0.19	0.001	8.00 **	0.024	1.37	0.004
Gender	1	2.20	0.007	0.05	0.000	0.05	0.000
Age	1	6.85 **	0.020	0.62	0.002	1.29	0.004
BMI							
Underweight vs Normal (A)	1	4.02 *	0.012	0.15	0.000	24.38 **	0.069
Obese vs No answer (B)	1	6.27 *	0.019	0.24	0.001	0.15	0.000
(A) vs (B)	1	0.04	0.000	0.09	0.000	13.81 **	0.040
Education							
Junior high school or less vs Senior high school (C)	1	0.02	0.000	0.73	0.002	1.34	0.004
University or higher vs No answer (D)	1	0.57	0.002	2.82	0.009	0.38	0.001
(C) vs (D)	1	0.86	0.003	1.58	0.005	1.58	0.002
Living alone							
Living alone vs Living with someone (E)	1	11.41 **	0.034	2.48	0.007	0.33	0.001
(E) vs No answer	1	0.79	0.002	0.06	0.000	1.70	0.005

* *p* < 0.05, ** *p* < 0.01, *η*^2^ Eta-squared, Adjusted *R*^2^: Isolation risk = 0.069, Fall risk = 0.041, Poor nutrition risk = 0.067.

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
