# Peer review of "Relationship between Perceived Indoor Temperature and Self-Reported Risk for Frailty among Community-Dwelling Older People"

_ijerph, 2019, doi:10.3390/ijerph16040613_

Round 1
Reviewer 1 Report
This is an interesting article about the relationships between frailty (the DV) and both perceived indoor temperature and economic satisfaction (main IVs). The results showed an interaction effect between the two IVs in predicting the DV, such that frailty was much higher for the group with both a cold indoor temperature and who were unsatisfied with their finances. This interaction effect was particularly significant for the frailty domain falls risk (but not for isolation risk or nutrition risk).
These relationships are interesting because they are not immediately obvious and they have public health implications.
I liked the Introduction and found the graphs helpful.
I have issues with the statistical analyses as reported and also have several English language edits to suggest.
Firstly, there is an issue with the numbers of participants cited in various tables. Why are there only 264 participants in Table 2 but 342 in other tables?
Other issues are more difficult to assess. The numbers of participants in the four groups were widely discrepant, with only 28 participants in the cold-satisfied group, compared with 200 in the warm-unsatisfied group. Where any statistical adjustments made to deal with these discrepancies?
Next, the contribution of the covariates is difficult to assess. Three covariates were categorical (Living arrangements) or ordinal (education and BMI) and there are missing values on all of these, especially on BMI and education. The degrees of freedom quoted in tables 3 and 4 are appropriate, but the role of these variables in the models is difficult to assess. Which groups had the highest frailty?
Further, strictly speaking, the covariates in an analysis of covariance should not be correlated with other IVs -- and this case it looks as if perception of indoor temperature could be related to at least three -- gender, economic satisfaction and living arrangements.
The covariates should, however, be related to the DV. Were all of the covariates related to the DV in bivariate analyses? Was the relationship between BMI and the DV linear or curvilinear?
Table 3 needs a better title. Presumably, this table sets out the results of the first analysis of covariance. It can be better to start with the DV (e.g., Analysis of total frailty score by perceived indoor temperature and participant characteristics).
The heading for table 4 is wrong, since predictors of total frailty score are not reported.
The Measures section of the article needs more detail. What thresholds were used to classify BMI? Also, the first paragraph of the Data analysis section should perhaps be moved to Measures.
English expression and typos
The English expression is good in most respects. The following are suggested edits.
Hyphenate community-dwelling
Change "that was accounting" to "that accounted'
What is meant by "different" on line 36? Do you mean various?
Replace "compared to" with "than"
On line 52, it would help the reader know what is meant by "old-fashioned attitudes" -- such as?
Replace "but there is not a single agreed upon definition" with "but there is no single agreed definition"
Delete the comma after [14]
Insert "and" before "female gender" on line 60.
One line 66, you must state whose frailty you are talking about -- the caregiver's or the care-recipient's?
Line 74 change "if" to "whether"
Line 86, change to "and 54 participants did not disclose their age".
In Table 1, change Q10 to "Do you avoid going outside because of fear of falling?
Change Q11 to "Have you been hospitalised in the past year? (Or use US spelling consistent with the rest of the article, hospitalized.)
Line 128, subscales should be plural.
Line 137 change "as" to "a"
Line 162, the referent is missing (than other groups)
Line 167-68, there is a problem with the syntax. Re-write: the people who both perceived a cold indoor winter temperature and were unsatisfied with their economic situation . . .
I think you should hyphenate poverty-risk-related
Line 176 relationships should be plural
Line 177, keep to the past tense, "when it came"
Line 191 use the plural "systems"
Line 194 change "less" to "lower"
Line 200, fix the punctuation
Line 201, it is not clear what you mean by "that". Are you saying that a larger proportion of winter fallers had lower limb weakness than those who fell in other seasons?
Line 208, change "of" to "for"
Line 215 insert "people" after "elderly"
Line 221 make investigations plural (consistent with what follows in this sentence)
Line 222. I think investigators should also be plural, hence the apostrophe is in the wrong place (write "trained investigators' support")
Line 224, change the comma to a semi-colon
Line 228, insert "the" before "economic status"
Line 234, insert a comma after "that" to set off the following phrase properly
Line 235, it is not good practice to use e.g. outside of brackets, and a preposition is missing. Write: "for example, by supplementing"
Line 236 delete "the" before "older"
Line 237, delete "the" before "society", change "of" to "for", and delete "the" before "public expenditure"
Author Response
15th February 2019
Dear reviewer,
On behalf of all the authors, I would like thank you for the time and effort you have dedicated to providing insightful feedback on ways to strengthen our paper. Thus, it is with great pleasure that we resubmit our article for further consideration. We have incorporated changes that reflect the detailed suggestions you have graciously provided. We also hope that our edits and responses provided below satisfactorily address all the issues and concerns you have noted.
To facilitate your review of our revisions, the following is a point-by-point response to the questions and comments delivered from you.
Major comments
Point 1: Firstly, there is an issue with the numbers of participants cited in various tables. Why are there only 264 participants in Table 2 but 342 in other tables?
Response 1: Our apologize that 264 was in Table 2. It was an error on our part and 342 is correct. We have corrected the table accordingly.
Point 2: Other issues are more difficult to assess. The numbers of participants in the four groups were widely discrepant, with only 28 participants in the cold-satisfied group, compared with 200 in the warm-unsatisfied group. Where any statistical adjustments made to deal with these discrepancies?
Response 2: The groups were determined based on the sample characteristics (on perceived temperature and economic satisfaction). We made the assumption that this is the distribution within our sample, and as equal group sizes are not necessary for the types of analyses we have chosen, we have not made any adjustments due to these different sizes.
Point 3: Next, the contribution of the covariates is difficult to assess. Three covariates were categorical (Living arrangements) or ordinal (education and BMI) and there are missing values on all of these, especially on BMI and education. The degrees of freedom quoted in tables 3 and 4 are appropriate, but the role of these variables in the models is difficult to assess. Which groups had the highest frailty?
Response 3: We agree that it could be useful to provide more details about the different levels of the categorical/ordinal covariates. We reconsidered the analysis and have shown new models of ANCOVA (Table 3) and MANCOVA (Table 4) including results for the different levels of categorical covariates. Figure 1 and 2 were also updated regarding to this change.
Point 4: Further, strictly speaking, the covariates in an analysis of covariance should not be correlated with other IVs -- and this case it looks as if perception of indoor temperature could be related to at least three -- gender, economic satisfaction and living arrangements.
Response 4: We used Spearman’s correlation analysis to check the correlation between each covariate and the independent variables. It confirmed that correlation coefficient in all pairs were not larger than 0.2. We added that information in the Data Analysis section.
Point 5: The covariates should, however, be related to the DV. Were all of the covariates related to the DV in bivariate analyses? Was the relationship between BMI and the DV linear or curvilinear?
Response 5: Yes, we agree with your point on the relationship between the covariates and the DV. We specifically chose these covariates because we know from previous research that they are related to frailty. Due to this known association from previous research, we do think it is necessary to present each bivariate relationship in the manuscript. Further, even if the bivariate correlations are not significant, the covariates can still have an impact on the overall model. We did look at the bivariate correlations for each covariate and each was related to at least one of the DV: total score of frailty, isolation risk, falls risk, or poor nutrition risk. BMI was related to frailty index and the relationship was linear that underweight had the highest risk of frailty.
Point 6: Table 3 needs a better title. Presumably, this table sets out the results of the first analysis of covariance. It can be better to start with the DV (e.g., Analysis of total frailty score by perceived indoor temperature and participant characteristics).
Response 6: Thank you for your suggestion. We changed the title for table 3 to “Analysis of covariance of total frailty score by perceived indoor temperature and participant characteristics”.
Point 7: The heading for table 4 is wrong, since predictors of total frailty score are not reported.
Response 7: Thank you for catching this error on our part. We changed the heading for table 4 in line with your previous comment.
Point 8: The Measures section of the article needs more detail. What thresholds were used to classify BMI? Also, the first paragraph of the Data analysis section should perhaps be moved to Measures.
Response 8: We added the thresholds of BMI into table 2 and moved the first paragraph of the Data analysis section to Measures.
English expression and typos
Point 9: What is meant by "different" on line 36? Do you mean various?
Response 9: Yes, we have changed it to various.
Point 10: Line 201, it is not clear what you mean by "that". Are you saying that a larger proportion of winter fallers had lower limb weakness than those who fell in other seasons?
Response 10: That was what we meant. We revised the sentence.
Other language suggestions and typos were corrected as recommended.
Again, thank you for giving us the opportunity to strengthen our manuscript with your valuable comments and queries. We have worked hard to incorporate your feedback and hope that these revisions persuade you to accept our submission.
Sincerely,
Yukie Nakajima, M.E., JSPS Research Fellow (DC)
School of Science for Open and Environmental Systems
Graduate School of Science and Technology
Keio University
3-14-1 Hiyoshi, Kohokuku, Yokohama 223-8522 Japan
+81 45 566 1770
yukie1222@z2.keio.jp

Reviewer 2 Report
This paper deals with the association between perceived indoor temperature and frailty risk. This paper is well written and deals with important issue among aging population. However, I have some concerns and suggestions before this paper is considered for publication.
Major issues
In materials and methods section, information on response rate is missing. Of the 473 survey participants, 131 were excluded. This is a large portion. Authors need to describe if there is any systematic difference, for example, in age or frailty status, between participants and non-participants. This kind of information is also necessary in limitation section.
In data analyses section, authors stated that MANCOVA was used to test the three subscales of the frailty index using the same independent variables and covariates as the previous model (ANCOVA). What are the independent variables? Please specify.
In above MANCOVA, did authors test the interaction of the three subscales? If so, what was the result?
In limitation section, please write more about the possibility of bias by using frail older persons (obviously they are not general older persons) in this study.
Minor issues
Usually, we use “p value” instead of “alpha” (line 131, p4)
Please use the term consistently to avoid confusion among readers. For example, “homebound” should be “isolation” in Table 2.
Description about the sign (η2 ) is needed in Table 3 and 4.
What does it mean “ * “ putted on “perceived indoor temperature” in Table 3?
Reconsider the title of the table 4. “Total frailty score” should be “three subscales of the frailty index”.
Author Response
15th February 2019
Dear reviewer,
On behalf of all the authors, I would like thank you for the time and effort you have dedicated to providing insightful feedback on ways to strengthen our paper. Thus, it is with great pleasure that we resubmit our article for further consideration. We have incorporated changes that reflect the detailed suggestions you have graciously provided. We also hope that our edits and responses provided below satisfactorily address all the issues and concerns you have noted.
To facilitate your review of our revisions, the following is a point-by-point response to the questions and comments delivered from you.
Major issues
Point 1: In materials and methods section, information on response rate is missing. Of the 473 survey participants, 131 were excluded. This is a large portion. Authors need to describe if there is any systematic difference, for example, in age or frailty status, between participants and non-participants. This kind of information is also necessary in limitation section.
Response 1: When we compared the characteristics between participants and non-participants, only poor nutrition risk which is one of the subscale of frailty was significantly different (p=.030), and participants (Mean score of poor nutrition risk=0.97, SD=0.96) had slightly higher risk of poor nutrition than non-participants (Mean score of poor nutrition risk=0.73, SD=0.86). Corresponding changes have been made in the manuscript.
Point 2: In data analyses section, authors stated that MANCOVA was used to test the three subscales of the frailty index using the same independent variables and covariates as the previous model (ANCOVA). What are the independent variables? Please specify.
Response 2: We added the information about independent variables in the Data analysis section.
Point 3: In above MANCOVA, did authors test the interaction of the three subscales? If so, what was the result?
Response 3: We find this question somewhat confusing as to what you mean by testing the interactions among the three dependent variables. Did you meant the associations among the dependent variables? If you did mean the associations among them, the MANCOVA takes the covariances among these three dependent variables into account within the analysis in the calculation of the linear composite of the three variables. If you meant something different, could you please clarify this question, and we will be happy to address it further.
Point 4: In limitation section, please write more about the possibility of bias by using frail older persons (obviously they are not general older persons) in this study.
Response 4: Thank you for providing these insights. We added some additional information about thermal sensitivity in the limitation section.
Minor issues
Point 5: What does it mean “ * “ putted on “perceived indoor temperature” in Table 3?
Response 5: “*” indicated that it is an interaction of perceived indoor temperature and economic satisfaction. We changed where the new line starts to make it more legible.
Again, thank you for giving us the opportunity to strengthen our manuscript with your valuable comments and queries. We have worked hard to incorporate your feedback and hope that these revisions persuade you to accept our submission.
Sincerely,
Yukie Nakajima, M.E., JSPS Research Fellow (DC)
School of Science for Open and Environmental Systems
Graduate School of Science and Technology
Keio University
3-14-1 Hiyoshi, Kohokuku, Yokohama 223-8522 Japan
+81 45 566 1770
yukie1222@z2.keio.jp
